# The RgaS-RgaR two-component system promotes *Clostridioides difficile* sporulation through a small RNA and the Agr1 system

**Adrianne N. Edwards**[ID]\*, **Shonna M. McBride**[ID]

Department of Microbiology and Immunology, Emory University School of Medicine, Emory Antibiotic Resistance Center, Atlanta, Georgia, United States of America

\* anehrli@emory.edu

## Abstract

The ability to form a dormant spore is essential for the survival of the anaerobic pathogen, *Clostridioides difficile*, outside of the mammalian gastrointestinal tract. The initiation of sporulation is governed by the master regulator of sporulation, Spo0A, which is activated by phosphorylation. Multiple sporulation factors control Spo0A phosphorylation; however, this regulatory pathway is not well defined in *C. difficile*. We discovered that RgaS and RgaR, a conserved orphan histidine kinase and orphan response regulator, function together as a cognate two-component regulatory system to directly activate transcription of several genes. One of these targets, *agrB1D1*, encodes gene products that synthesize and export a small quorum-sensing peptide, AgrD1, which positively influences expression of early sporulation genes. Another target, a small regulatory RNA now known as SpoZ, impacts later stages of sporulation through a small hypothetical protein and an additional, unknown regulatory mechanism(s). Unlike Agr systems in many organisms, AgrD1 does not activate the RgaS-RgaR two-component system, and thus, is not responsible for autoregulating its own production. Altogether, we demonstrate that *C. difficile* utilizes a conserved two-component system that is uncoupled from quorum-sensing to promote sporulation through two distinct regulatory pathways.

## Author summary

The formation of an inactive spore by the anaerobic, gastrointestinal pathogen, *Clostridioides difficile*, is required for its survival outside of the mammalian host. The sporulation process is induced by the regulator, Spo0A; yet, how Spo0A is activated in *C. difficile* remains unknown. To address this question, we investigated potential activators of Spo0A. Here, we demonstrate that the sensor RgaS activates sporulation, but not by direct activation of Spo0A. Instead, RgaS activates the response regulator, RgaR, which in turn activates transcription of several genes. We found two direct RgaS-RgaR targets independently promote sporulation: *agrB1D1*, encoding a quorum-sensing peptide, AgrD1, and *spoZ*, encoding a small regulatory RNA. Unlike most other characterized Agr systems, the AgrD1 peptide does not affect RgaS-RgaR activity, indicating that AgrD1 does not activate

**Data Availability Statement:** All relevant data are within the manuscript and its Supporting Information files, and genome and transcriptomic sequence files were deposited to the NCBI

Sequence Read Archive (SRA) BioProject
PRJNA986905.

**Funding:** This research was supported by the U.S.
National Institutes of Health through research
grants AI116933 and AI156052 to S.M.M. The
funding agency had no role in the design or
execution of this study, the analysis of the data, or
the decision to publish the results.

**Competing interests:** The authors have declared
that no competing interests exist.

its own production through RgaS-RgaR. Altogether, the RgaS-RgaR regulon functions at multiple points within the sporulation pathway to tightly control *C. difficile* spore formation.

## Introduction

*Clostridioides difficile* is an anaerobic gastrointestinal pathogen that causes severe diarrheal disease [1]. The vegetative cells of this strict anaerobe undergo a dramatic morphogenesis within the mammalian gastrointestinal tract to form dormant, highly resistant endospores. This infectious spore enables long-term survival outside of the host and facilitates transmission to new hosts [2]. The initiation of spore formation requires the activation of the highly conserved transcriptional regulator, Spo0A, which is encoded in all endospore-forming bacteria [2–4]. Spo0A DNA-binding activity is controlled by the phosphorylation of a conserved aspartate residue [5,6], allowing phosphorylated Spo0A (Spo0A~P) to directly activate early sporulation-specific gene expression and triggering the entry into sporulation [5–8].

In other well-studied spore formers, most notably the soil dweller *Bacillus subtilis*, Spo0A phosphorylation is tightly controlled by an expansive regulatory network comprised of kinases, phosphatases, phosphotransferases, and additional regulators [9]. *B. subtilis* Spo0A phosphorylation is achieved via a phosphorelay that begins with sensor histidine kinases that autophosphorylate and transmit a phosphoryl group through a response regulator (Spo0F) and a phosphotransferase (Spo0B) to Spo0A. Many of the key regulatory proteins that govern Spo0A activation in *B. subtilis* are not well conserved or are absent in *C. difficile* genome [10–13], suggesting that *C. difficile* utilizes unique factors and regulatory pathways to initiate spore formation. Three *C. difficile* phosphotransfer proteins (PtpA, PtpB, and PtpC), originally annotated as orphan histidine kinases, appear to inactivate Spo0A under the conditions tested [14,15]. Thus, the primary driver of *C. difficile* Spo0A phosphorylation and activation is unknown.

Environmental cues and nutrient availability influence *C. difficile* sporulation initiation in the host gastrointestinal tract [16–20]. Additionally, bacterial global signaling systems play a significant role in the decision to initiate sporulation. The intracellular nucleotide second messenger, c-di-GMP, is an inhibitor of early sporulation in *C. difficile* [21]. In other spore formers, quorum-sensing systems are well established for regulating sporulation initiation [22,23]. The RRNPP family of multifunctional proteins in *B. subtilis* and *Bacillus cereus* sp. directly bind small quorum-sensing peptides to regulate their activity, which impacts Spo0A phosphorylation [24,25]. The *C. difficile* RRNPP ortholog, RstA, positively influences spore formation by directly binding an inhibitor of sporulation [26,27]. However, RstA differs from most members of the RRNPP family in that its cognate quorum-sensing peptide is not adjacently encoded on the genome and has yet to be discovered [26], providing additional evidence that *C. difficile* employs distinct regulatory pathways to initiate sporulation.

Another conserved quorum sensing system implicated in spore formation is the accessory gene regulator (Agr) system, which utilizes an extracellular, cyclic autoinducing peptide (also generically known as the AIP) as the quorum-sensing signal. The prototypical Agr system encompasses the autoinducing peptide (AgrD), a transmembrane protease (AgrB) that processes and exports AgrD, a sensor histidine kinase (AgrC), which directly senses extracellular AgrD, and the cognate response regulator (AgrA) [28]. Prior studies in *Clostridium botulinum*, *Clostridium sporogenes*, and *Clostridium perfringens* demonstrated that disruption or knockdown of the *agrBD* genes resulted in decreased spore formation [29,30]. The majority of *C. difficile* genomes encode an *agrB1D1* operon with no apparent cognate *agrA1* or *agrC1* present

[29,31–33]. Recent work revealed that the *agrB1D1* locus positively impacts early sporulation gene expression and spore formation [34]; however, the identity of the AgrD1 receptor remains elusive.

*C. difficile* encodes an orphan histidine kinase (CD0576; herein, RgaS) that is annotated as an ArgC-like sensor histidine kinase. Because the sporulation factor(s) that activate Spo0A are unclear, we asked if RgaS plays a role in *C. difficile* sporulation initiation. Our results reveal that RgaS signals through an orphan response regulator, RgaR, to promote *C. difficile* spore formation. We demonstrate that the conserved residues required for phosphoryl group transfer in both RgaS and RgaR are necessary for their function as a cognate two-component system. Further, RgaSR directly activate transcription of several loci, including *agrB1D1* and a small regulatory RNA, *n00620* (*nc063*; herein referred to as *spoZ*). We establish that RgaSR influences spore formation through direct regulation of both *agrB1D1* and *spoZ* transcription. Additionally, we show that the effects of AgrD1 and *spoZ* are independent and impact different stages of sporulation. Unlike prototypical Agr systems, RgaS does not respond to the AgrD1 peptide, indicating that AgrB1D1 has adapted distinct functions that are independent of the RgaSR two-component system. Finally, we present evidence that RgaSR promotion of spore formation through AgrD1 and SpoZ is functionally conserved in diverse *C. difficile* strains.

## Results

### The orphan histidine kinase RgaS and the RgaR response regulator promote *C. difficile* sporulation

The master response regulator of sporulation, Spo0A, is activated by phosphorylation. Multiple kinases activate sporulation in other spore-forming organisms; however, these factors are not conserved in *C. difficile*. Although three phosphotransfer proteins are proposed to act as phosphatases to inhibit Spo0A activity [15], a factor that directly phosphorylates Spo0A has not been discovered [13]. We sought to determine if other encoded kinases could activate Spo0A to drive sporulation. We identified an additional, putative orphan histidine kinase within the *C. difficile* genome, *rgaS* (*CD0576*), that has limited similarity to the other sporulation-associated phosphotransfer proteins, including in the dimerization and histidine autophosphorylation domain (H-box; **S1 Fig**). To examine whether RgaS contributes to *C. difficile* sporulation, we first utilized CRISPR interference (CRISPRi) to directly repress *rgaS* gene expression. As the addition of xylose to sporulation medium reduces sporulation frequency [15], we modified the *C. difficile* CRISPRi plasmid [35] to drive expression of the nuclease-deactivated version of the *dCas9* gene with the nisin-inducible *cprA* promoter. A scrambled sgRNA (sgRNA-*neg*) was included as a control. We assessed the sporulation frequency of 630Δ*erm* expressing either sgRNA-*neg* or sgRNA-*rgaS* by determining the ratio of spores to vegetative cells after 24 h of growth on 70:30 sporulation agar, with and without nisin. Expressing the anti-*rgaS* target resulted in a ~6-fold decrease in spore formation compared to the control strain (**Fig 1A**). To ensure that *rgaS* was directly targeted, *rgaS* transcripts were measured by quantitative reverse-transcription PCR (qRT-PCR), and as expected, *rgaS* transcripts were decreased ~25-fold in the presence of nisin (**S2A Fig**). This data suggests that RgaS positively impacts *C. difficile* sporulation.

RgaS possesses a histidine kinase-like ATPase domain that is similar to the AgrC histidine kinase of *Staphylococcus aureus* (**Fig 1B**). Additional AgrC-like kinases have been identified in *Clostridia*, including the *Clostridium perfringens* VirS [36] and the *C. difficile* AgrC2, which is encoded only in the epidemic ribotype 027 strains [37]. Recent work established that *C. perfringens* VirS, which directly phosphorylates its cognate response regulator, VirR [38], responds to the Agr quorum sensing peptide produced by *C. perfringens* to activate VirR-

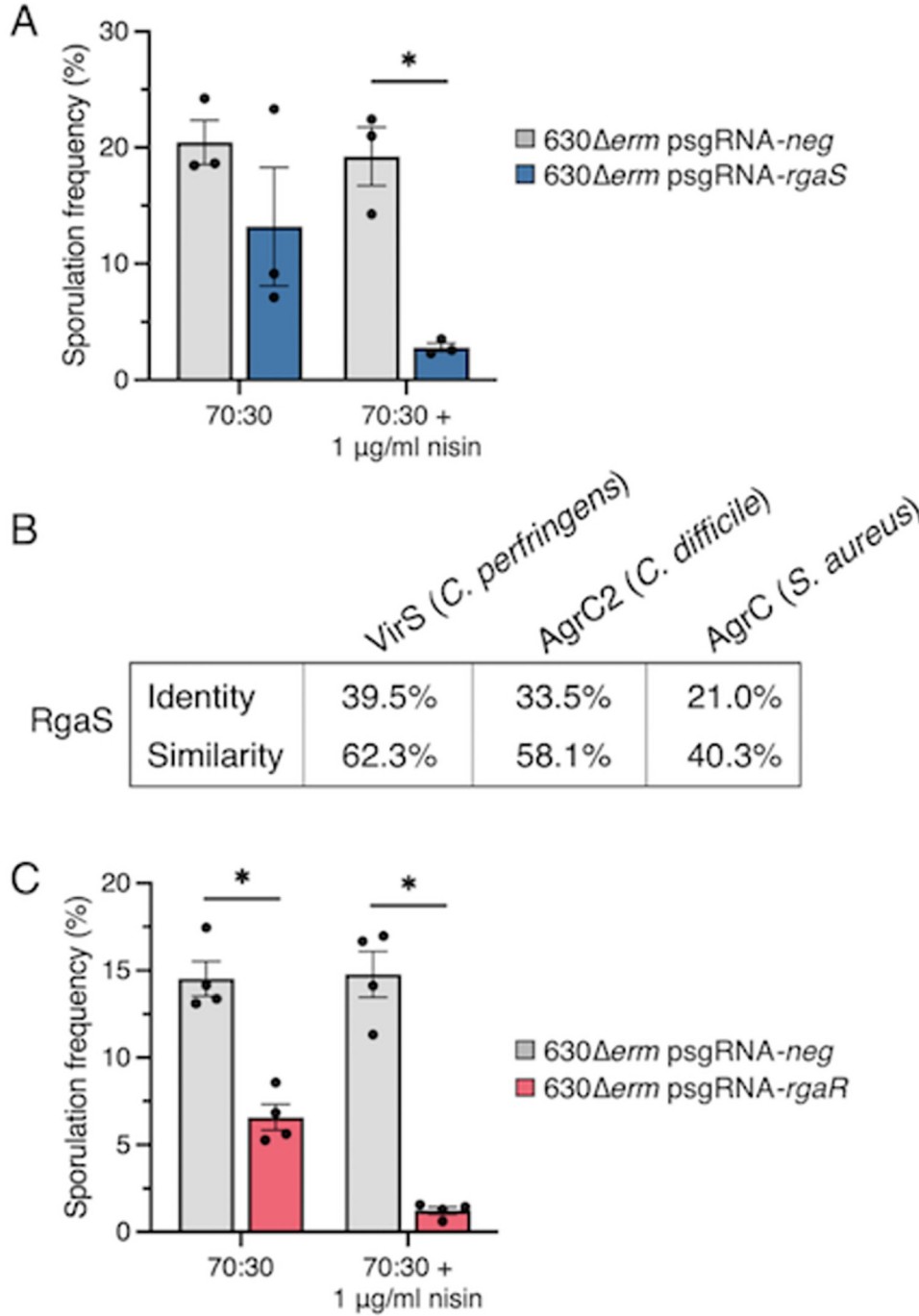

**Fig 1. CRISPRi knockdown of *rgaS* (*CD0576*) and *rgaR* (*CD3255*) results in decreased sporulation frequency. (A**, **C**) Ethanol-resistant spore formation at $H_{24}$ in 630Δ*erm* psgRNA-*neg* (MC2065), 630Δ*erm* psgRNA-*rgaS* (MC2066), and 630Δ*erm* psgRNA-*rgaR* (MC2227) grown on 70:30 agar supplemented with 2 μg/ml thiamphenicol and 1 μg/m nisin, as indicated. The means and standard deviation of three independent biological replicates (A) or standard error of the means of four independent biological replicates (C) are shown. *, P < 0.01 by a Student's *t*-test. (**B**) Percent identity and similarity of *Clostridium perfringens* VirS, *C. difficile* AgrC2 (CDR20291_3188), and *Staphylococcus aureus* AgrC to the histidine kinase RgaS in *C. difficile* 630.

dependent gene expression [39]. RgaS shares high similarity and identity with *C. perfringens* VirS and *C. difficile* R20291 AgrC2 (**Fig 1B**), suggesting that RgaS functions as a histidine kinase that directly senses an extracellular quorum sensing peptide.

We hypothesized that if RgaS is a VirS-like histidine kinase, then its cognate response regulator would be similar to *C. perfringens* VirR. We scanned the *C. difficile* 630 genome for orphan response regulators and located two VirR orthologs: CD1089 (RgbR) and CD3255 (RgaR). A prior investigation showed that overexpression of *rgaR*, but not *rgbR*, in a *C. perfringens virR* mutant partially complemented toxin production, suggesting that RgaR functions as a VirR ortholog [40]. Further, they discovered that RgaR directly activated transcription of three *C. difficile* operons, including the partial *agrB1D1* locus, while an *rgbR* mutant did not exhibit any transcriptional changes [40]. To determine whether RgaR affects *C. difficile* sporulation, we knocked down *rgaR* gene expression by CRISPRi. *C. difficile* sporulation was significantly decreased (~12-fold) in the 630Δ*erm* strain expressing sgRNA-*rgaR* compared to the parental control, indicating that RgaR positively influences spore formation (**Fig 1C**). *rgaR* transcript levels were decreased ~4-fold (**S2B Fig**), confirming that the *rgaR* locus was directly targeted by CRISPRi.

To further explore the impact of RgaS and RgaR on *C. difficile* sporulation, we generated *rgaS* and *rgaR* deletion mutants using allele-coupled exchange [41] (see Methods; **S3 Fig**). Sporulation phenotypes of the mutants were assessed after 24 h growth on sporulation agar. The *rgaS* and *rgaR* single mutants sporulated at lower frequencies (~7%) than the 630Δ*erm* parent (~21%; **Fig 2A**), corroborating the CRISPRi knockdown results. These sporulation phenotypes were also confirmed by phase contrast microscopy (**Fig 2B**), as fewer phase bright spores are visible in the *rgaS* and *rgaR* mutants. We also created a double *rgaS rgaR* mutant and found that the double mutant exhibited similar sporulation frequencies as the single *rgaS* and *rgaR* mutants (**Fig 2C**), suggesting that RgaS and RgaR function in the same regulatory pathway to affect sporulation.

To ensure that the decreased sporulation frequencies of the *rgaS* and *rgaR* mutants are solely the result of the deleted genes, we complemented these mutants by expressing the wild-type *rgaS* and *rgaR* alleles from their native promoters on the chromosome via the Tn*916* conjugative transposon. Expression of the *rgaS* and *rgaR* alleles in their respective mutants restored sporulation frequencies to greater than wild-type levels (**Fig 2C**).

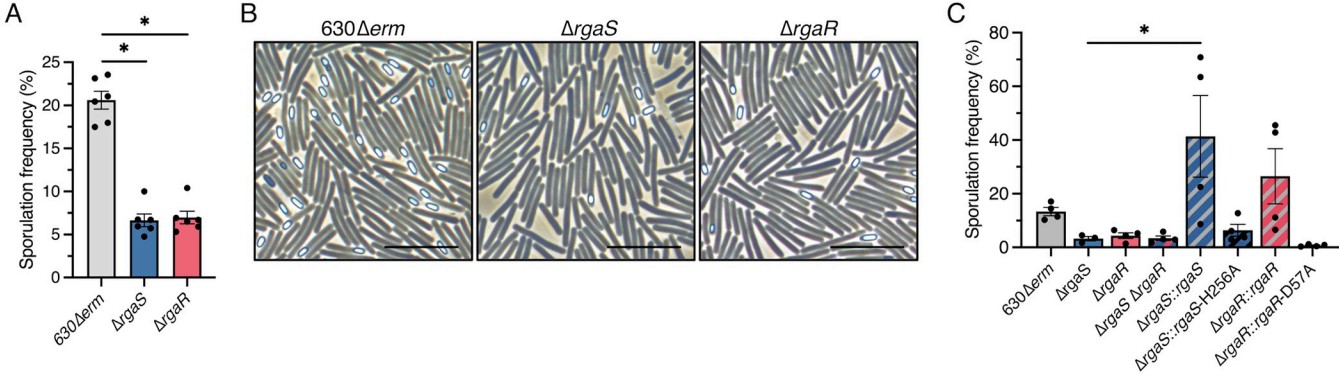

**Fig 2. *C. difficile rgaS* and *rgaR* mutants exhibit low sporulation frequencies and require their conserved histidine or aspartate residue for function. (A)** Ethanol-resistant spore formation and (**B**) representative phase-contrast micrographs at H₂₄ in 630Δ*erm*, 630Δ*erm* Δ*rgaS* (MC2228), and 630Δ*erm* Δ*rgaR* (MC2229) grown on 70:30 agar. Scale bar denotes 10 μm. (**C**) Ethanol-resistant spore formation at H₂₄ of 630Δ*erm*, 630Δ*erm* Δ*rgaS* (MC2228), 630Δ*erm* Δ*rgaR* (MC2229), 630Δ*erm* Δ*rgaS* Δ*rgaR* (MC2236), 630Δ*erm* Δ*rgaS* Tn*916*::*rgaS* (MC2278), 630Δ*erm* Δ*rgaS* Tn*916*::*rgaS*-H256A (MC2318), 630Δ*erm* Δ*rgaR* Tn*916*::*rgaR* (MC2319), and 630Δ*erm* Δ*rgaR* Tn*916*::*rgaR*-D57A (MC2320) grown on 70:30 agar. The means and standard error of the means of at least four independent biological replicates are shown. *, P < 0.01 by a one-way ANOVA followed by Dunnett's multiple comparisons test (A) or Tukey's multiple comparisons test (C).

Our data suggest that RgaS and RgaR function in the same regulatory pathway as a two-component regulatory system (TCS), where the RgaS histidine kinase undergoes autophosphorylation at the conserved histidine residue and transfers the phosphoryl group to the conserved aspartate residue of the RgaR response regulator [42]. To determine whether the predicted functional residues of RgaS and RgaR are essential for their activity, we replaced the conserved histidine residue of RgaS and the conserved aspartate residue of RgaR with alanine, using site-directed mutagenesis, and expressed these alleles in their respective mutants via the Tn*916* conjugative transposon. Neither the *rgaS*-H256A nor the *rgaR*-D57A allele complemented the low sporulation phenotypes of the respective mutants (**Fig 2C**), demonstrating that these residues are required for RgaS and RgaR to promote sporulation.

## Late-stage sporulation is downregulated in the *rgaR* mutant

To better understand how the RgaS-RgaR TCS promotes sporulation, we performed RNA-seq on the *rgaR* mutant during growth on sporulation agar ($H_{12}$) to identify differentially expressed genes during sporulation (**S1 Table**). The development of spores requires the activation of Spo0A followed by the sequential activation of the sporulation sigma factors, SigF, SigE, SigG, and SigK [10,43]. RNA-seq analysis revealed that only a few Spo0A- or SigF-dependent transcripts were significantly decreased in the *rgaR* mutant (~22% of the Spo0A regulon and ~33% of the SigF regulon; **Table 1**), suggesting that RgaR does not strongly influence the initiation of sporulation. To confirm that Spo0A activity is not impacted in the *rgaS* and *rgaR* mutants, we measured transcript levels of *sigE*, a Spo0A~P-dependent gene, by qRT-PCR (**S4A Fig**). There were no significant differences in *sigE* transcript levels between the *rgaS*, *rgaR*, and parent strains, confirming that early sporulation gene regulation is not considerably impacted by RgaSR. In contrast, ~84% of the SigE regulon, 94% of the SigG, and 97% of the SigK regulons were significantly downregulated in the *rgaR* mutant (**Table 1**), revealing that RgaR promotes the progression to later stages of spore formation.

The RNA-seq results also revealed that expression of the toxin genes *tcdA* and *tcdB* were largely unaffected in the *rgaR* mutant, with a modest ~2-fold decrease in *tcdB* transcripts (**S1 Table**). We further assessed toxin production by measuring the total TcdA and TcdB toxin present in the supernatants of the *rgaS* and *rgaR* mutants. Toxin levels were unchanged between the *rgaS* mutant, the *rgaR* mutant, and the parent strain (**S4B Fig**). Altogether, these data confirm that the RgaSR TCS does not substantially impact toxin production in the conditions tested and suggests that the effects of RgaSR on sporulation regulation are independent of toxin regulation.

**Table 1. Late-stage sporulation genes are significantly downregulated in the *rgaR* mutant.**

| Regulon[a] | Upregulated genes | Unaffected genes | Downregulated genes | Total genes in regulon[b] | Percent of genes downregulated[c] |
|---|---|---|---|---|---|
| Spo0A-dependent | 12 | 56 | 19 | 87 | 22% |
| SigF-dependent | 0 | 14 | 7 | 21 | 33% |
| SigE-dependent | 0 | 17 | 89 | 106 | 84% |
| SigG-dependent | 0 | 2 | 31 | 33 | 94% |
| SigK-dependent | 0 | 1 | 29 | 30 | 97% |

[a] The Spo0A and sporulation-associated sigma factor regulons are based on those reported in [7].

[b] Genes whose expression were dependent on multiple sporulation-specific sigma factors were excluded from this analysis.

[c] These values are based on genes differentially expressed at least 2-fold and with $p < 0.05$ from the RNA-seq analyses of 630Δ*erm* Δ*rgaR* (MC2229) compared to the 630Δ*erm* parent strain grown on 70:30 agar at $H_{12}$.

## RgaS and RgaR have the same effects on gene expression

Prior work revealed that RgaR directly binds and activates transcription from three promoters that contain highly conserved VirR-like consensus binding sites [40]. These direct RgaR targets include the promoters for *agrB1D1*, the hypothetical factor *CD2098*, and *CD0587-CD0588*, a two-gene operon encoding two hypothetical proteins. Bioinformatical mining uncovered two additional VirR-like binding sites located in intergenic regions within the *C. difficile* genome [40], which have since been annotated. One VirR-like binding site is located upstream of a small ORF encoding a hypothetical protein, CD15111, and the second site lies immediately upstream of a predicted small regulatory RNA we have dubbed SpoZ (**Fig 3A**). Notably, the RNA-seq data revealed that only sporulation genes were downregulated in the *rgaR* mutant with the exception of these known direct RgaR targets (**S1 Table**), further supporting that this list encompasses all genes directly regulated by RgaR.

We next asked if these direct RgaR targets are similarly regulated in the *rgaS* mutant. Using qRT-PCR, we assessed transcript levels of RgaR-regulated genes in sporulating cells and

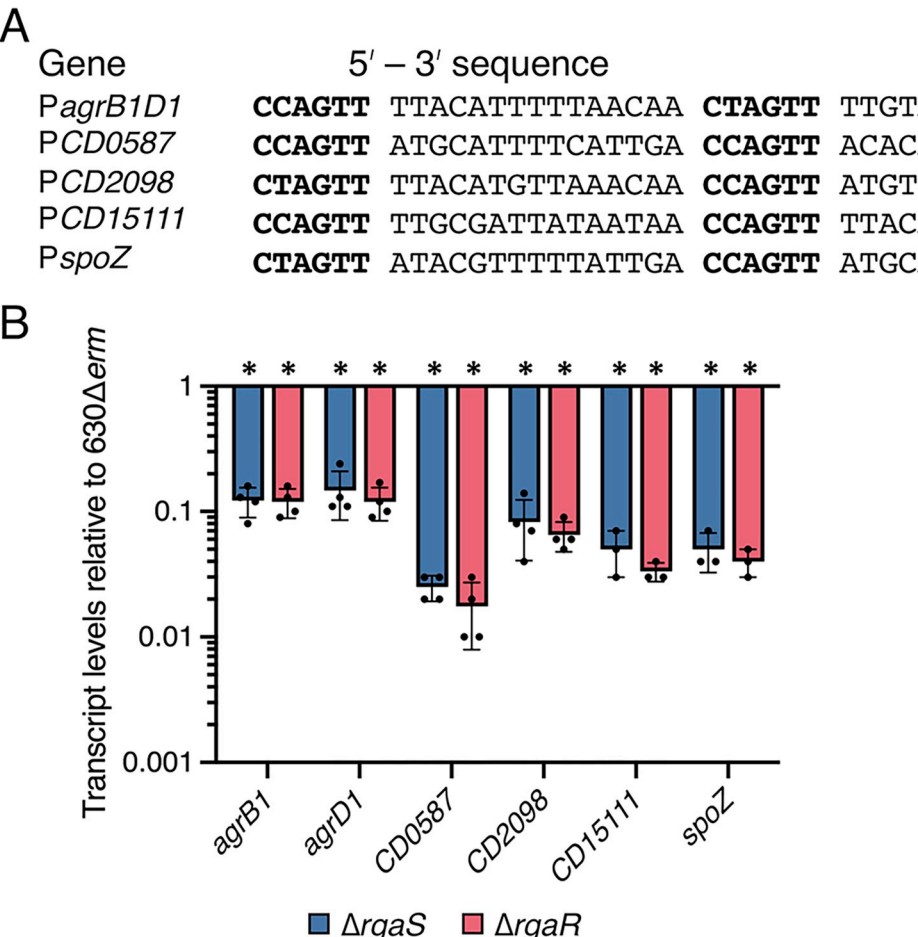

**Fig 3. RgaR-dependent gene expression is similarly decreased in both the *rgaS* and *rgaR* mutants.** (**A**) Alignment of the putative RgaR binding sites with the promoters of confirmed and predicted direct RgaR targets. The direct repeats are in bold. (**B**) qRT-PCR analysis of RgaR-dependent transcripts at $H_{12}$ in 630$\Delta erm$, 630$\Delta erm$ $\Delta rgaS$ (MC2228), and 630$\Delta erm$ $\Delta rgaR$ (MC2229) grown on 70:30 agar. The means and standard error of the means of at least three independent biological replicates are shown. *, P < 0.0001 by a one-way ANOVA followed by Dunnett's multiple comparisons test.

observed the same decrease in transcript levels for the *rgaS* and *rgaR* mutants (**Fig 3B**). Altogether, these data strongly support that RgaS is the cognate histidine kinase of the RgaR response regulator and that these proteins comprise a functional two-component regulatory system.

## RgaR regulates sporulation through the small regulatory RNA, SpoZ

To determine which direct RgaR target is responsible for the low sporulation phenotype in the *rgaS* and *rgaR* mutants, we used CRISPRi to individually knockdown transcription of *CD0587*, *CD2098*, *CD15111*, and *spoZ*, and assessed the impacts on sporulation. Knockdown of *CD0587*, *CD2098*, or *CD15111* did not alter sporulation frequency compared to the control (**Fig 4A**). However, the knockdown of *spoZ* significantly increased spore formation (**Fig 4A**). This was a surprising result given that the sporulation phenotype of the *spoZ* knockdown is opposite of what we observe for the *rgaS* and *rgaR* mutants. However, *spoZ* is the only known RgaR target that differentially impacts sporulation when targeted by CRISPRi, suggesting that the RgaS and RgaR influence sporulation frequency through the SpoZ small regulatory RNA. As before, we confirmed direct targeting of transcription by CRISPRi using qRT-PCR (**S5 Fig**).

Because the *spoZ* locus was not annotated when the original RgaR regulon was elucidated by bioinformatics, RgaR-dependent regulation of this locus had not been confirmed [40]. We created a reporter fusion of the *spoZ* promoter region to the alkaline phosphatase (AP) gene, *phoZ* [44]. We expressed P*spoZ::phoZ* in 630Δ*erm* and the *rgaR* mutant and found that AP activity was virtually eliminated in the *rgaR* mutant compared to the parent strain (**Fig 4B**),

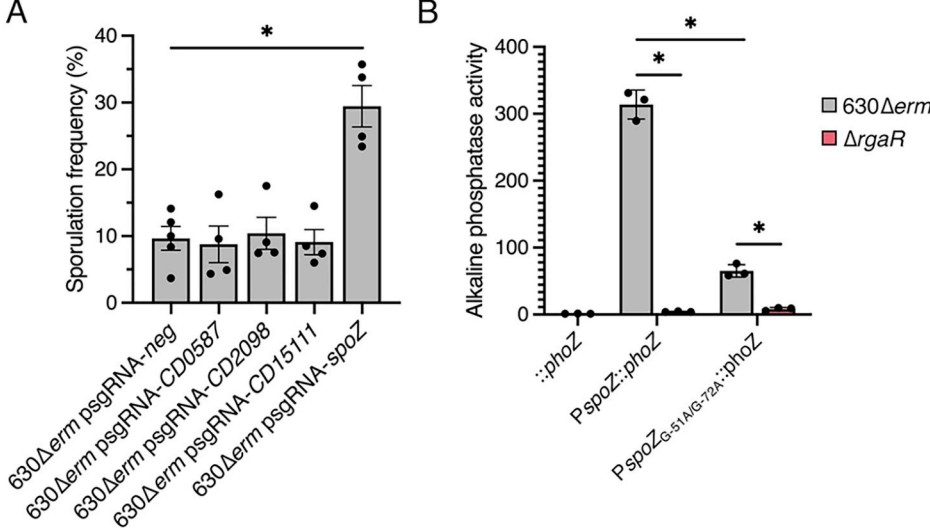

**Fig 4. CRISPRi knockdown of RgaR-dependent genes reveal that RgaR impacts sporulation through the regulatory sRNA, SpoZ (encoded by *CD630_n00620*).** (**A**) Ethanol-resistant spore formation of 630Δ*erm* expressing p*sgRNA-neg* (MC2065) or CRISPRi targets for *CD0587* (MC2269), *CD2098* (MC2270), *CD15111* (MC2271), or *spoZ* (MC2272) grown on 70:30 agar supplemented with 2 μg/ml thiamphenicol and 1 μg/m nisin at $H_{24}$. The means and standard errors of the means for four biological replicates are shown. (**B**) Alkaline phosphatase activity of either the wildtype P*spoZ::phoZ* or a site-directed mutagenized P*spoZ*$_{G-51A/G-72A}$*::phoZ* reporter fusion expressed from a plasmid in 630Δ*erm* (MC2329 and MC2357, respectively) or 630Δ*erm* Δ*rgaR* (MC2330 and MC2358, respectively). The promoterless *phoZ* reporter carried by 630Δ*erm* (MC448) was included as a negative control. The means and standard deviations for at least three biological replicates are shown. \*, $P < 0.01$ by one-way ANOVA followed by Dunnett's multiple comparisons test (**A**) or Tukey's multiple comparisons test (**B**).

indicating that RgaR is required for efficient *spoZ* expression. To confirm that the identified VirR/RgaR direct repeats are responsible for direct RgaR-dependent transcription of *spoZ*, we mutated the critical "G" residue within each direct repeat of the P*spoZ*::*phoZ* reporter. AP activity from the P*spoZ*$_{G-51A/G-72A}$::*phoZ* reporter was reduced ~5-fold in the *rgaR* mutant compared to the parent strain (**Fig 4B**), indicating that these two nucleotides within the direct repeats are necessary for direct RgaR binding to the *spoZ* promoter. Finally, AP activity from the P*spoZ*$_{G-51A/G-72A}$::*phoZ* reporter was reduced even further in the *rgaR* background, suggesting that RgaR still has some affinity to the mutated *spoZ* promoter region. Altogether, these data affirm that RgaR directly activates transcription from the *spoZ* promoter.

## The regulatory sRNA SpoZ influences *C. difficile* sporulation through multiple regulatory pathways

To determine how SpoZ impacts *C. difficile* sporulation, we first characterized the *spoZ* locus. Immediately downstream from the *spoZ* locus is a small ORF (58 amino acids), CD16671, which is encoded on the antisense strand (**Fig 5A**). To confirm the expression and determine the length of the *spoZ* transcript, we performed near-infrared northern blotting and detected a transcript corresponding to approximately 560 nt in the 630Δ*erm* background (**Fig 5B**). A similarly sized transcript, although less abundant, was observed in the R20291 background. No corresponding *spoZ* transcript was detected in the *rgaR* mutant, which was expected as RgaR is required for efficient activation of *spoZ* transcription. We further analyzed the cDNA synthesized from this locus in the parent strain (**S6A**, **S6B Fig**), and these data confirmed that the *spoZ* transcript terminates within the *CD16671* promoter region (**S6B Fig**). Altogether, these data corroborate the previously mapped transcription termination site for *spoZ*, slightly beyond the *CD16671* ORF (528 nt) [45], indicating that *spoZ* is encoded antisense to the entire CD16671 ORF. The predicted secondary structure of SpoZ was determined using UNAFold [46] and is shown in **S6C Fig**.

SpoZ was originally identified as a non-coding intergenic RNA and is predicted to act as an antisense target for the mRNA transcript of the neighboring ORF, CD16671 [45,47,48]. Subsequent studies discovered that SpoZ copurifies with Hfq, an RNA-binding protein that mediates post-transcriptional RNA interactions [48]. We hypothesized that SpoZ functions as an antisense RNA that targets the *CD16671* transcript for Hfq-dependent turnover. We assessed *CD16671* transcript levels in the *rgaS* and *rgaR* mutants and found that the *CD16671* transcript is significantly increased in these mutants (**Fig 5C**). These results suggest that SpoZ may promote sporulation by inhibiting accumulation of the *CD16671* transcript and subsequent protein product.

To test this hypothesis, we generated deletion mutants of either the entire *spoZ-CD16671* locus or only *CD16671* (see Methods; **Figs 5B**; **S7A and S7B**) and assessed their sporulation phenotypes. Like the *spoZ* knockdown strain, deletion of *spoZ-CD16671* resulted in increased sporulation frequency (~2.7-fold) compared to the parent strain (**Fig 5D**). However, deletion of *CD16671* did not impact sporulation frequency (**Fig 5D**). Because the decreased sporulation phenotype of the *rgaS* and *rgaR* mutants coincided with increased *CD16671* transcript levels, it is possible that the absence of CD16671 does not affect *C. difficile* spore formation but rather the accumulation of CD16671 inhibits sporulation. To explore this possibility, we overexpressed *CD16671* from an inducible promoter in the 630Δ*erm* background and found that sporulation frequency was reduced (~1.8-fold; **Fig 5E**). *CD16671* transcript levels were increased ~10-fold over wild-type levels (**S7C Fig**), suggesting that accumulation of CD16671 prevents the formation of spores. We also measured transcript levels of *sigE* and *sigG*, encoding sporulation-specific sigma factors, and found *sigE* and *sigG* transcript levels were also

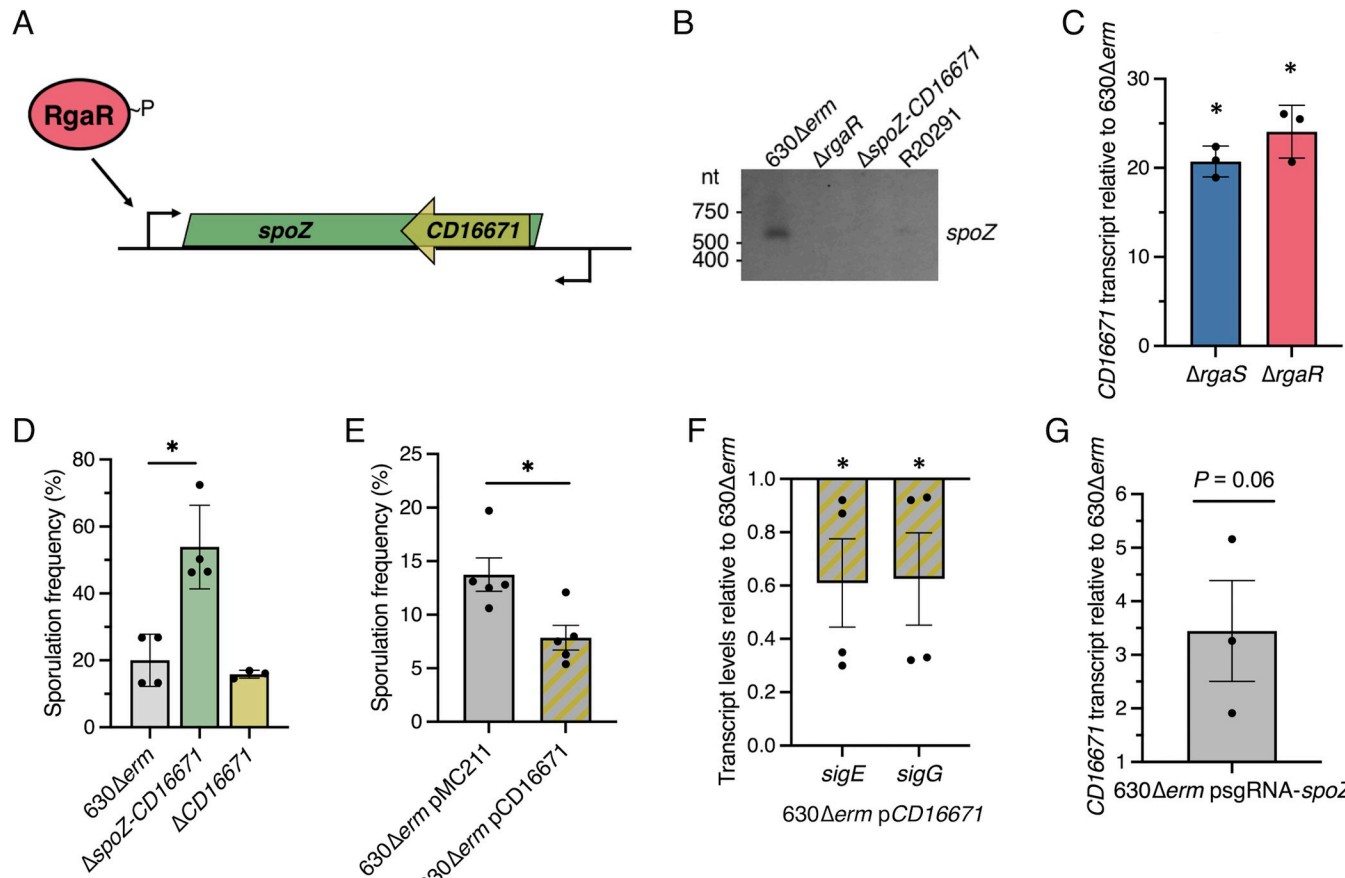

**Fig 5. The regulatory sRNA SpoZ influences *C. difficile* sporulation through multiple regulatory pathways.** (**A**) Genomic context of the sRNA, *spoZ*, in relation to the small ORF, *CD16671*. (**B**) Representative irNorthern blot of *spoZ* in 630Δ*erm*, 630Δ*erm* Δ*rgaR* (MC2229), 630Δ*erm* Δ*spoZ-CD16671* (MC2351), and R20291 grown on 70:30 agar at H$_{12}$. (**C**) qRT-PCR analysis of *CD16671* transcript levels in 630Δ*erm*, 630Δ*erm* Δ*rgaS* (MC2228), and 630Δ*erm* Δ*rgaR* (MC2229) grown on 70:30 agar at H$_{12}$. (**D**) Ethanol-resistant spore formation of 630Δ*erm*, 630Δ*erm* Δ*spoZ-CD16671* (MC2351), and 630Δ*erm* Δ*CD16671* (MC2363) grown on 70:30 agar at H$_{24}$. (**E**) Ethanol-resistant spore formation at H$_{24}$ of and (**F**) qRT-PCR analysis at H$_7$ of *sigE* and *sigG* transcripts in 630Δ*erm* pMC211 (MC282) and 630Δ*erm* p*CD16671* (MC2561) grown on 70:30 agar supplemented with 2 μg/ml thiamphenicol and 1 μg/ml nisin. (**G**) *CD16671* in 630Δ*erm* expressing psgRNA-*neg* (MC2065) or psgRNA-*spoZ* (MC2272) grown on 70:30 agar supplemented with 2 μg/ml thiamphenicol and 1 μg/m nisin. The means and standard deviation of three independent biological replicates (C, G) or standard error of the means of four independent biological replicates (D-F) are shown. *, P < 0.01 by a Student's *t*-test (E-G) or by a one-way ANOVA followed by a Dunnett's multiple comparisons test (C, D).

decreased ~1.6-fold when *CD16671* was overexpressed (**Fig 5F**), Altogether, this data indicates that increased CD16671 abundance inhibits *C. difficile* sporulation-specific gene expression and spore formation.

Because the *rgaS* and *rgaR* mutants have fewer *spoZ* transcripts and lower sporulation frequencies, and the *spoZ-CD16671* mutant and *spoZ* knockdown strains also have fewer *spoZ* transcripts but hypersporulate, we asked if the difference in the sporulation phenotypes between these strains is due to differences in *CD16671* transcript levels. We revisited the *spoZ* CRISPRi knockdown strain, which hypersporulated (**Fig 4A**), and discovered that *CD16671* transcript levels were only modestly increased (~3.4-fold; *P* = 0.06; **Fig 5G**), in contrast to the significant increase of *CD16671* transcript levels in the *rgaS* and *rgaR* mutants (~20-fold; **Fig 5C**) and the *CD16671* overexpression strain (~10-fold; **S7C Fig**). These data support the hypothesis that CD16671 accumulation inhibits sporulation, and provides additional evidence that SpoZ has an additional unidentified target(s) that mediates its effects on sporulation when CD16671 is not expressed at high levels.

Interestingly, 630Δ*erm* encodes a CD16671 ortholog, CD19903, which shares 80% nucleotide identity and 81% amino acid identity (**S8A Fig**). We asked whether SpoZ also affects *CD19903* transcript levels and found that there was no significant difference in *CD19903* transcript levels in the *rgaR* mutant (**S8B Fig**). We observed a statistically significant, but highly variable, ~1.8-fold increase in *CD19903* transcript levels in the *spoZ-CD16671* mutant. These data indicate that SpoZ does not influence *CD19903* in the conditions tested. It is likely that CD19903 functions similarly to CD16671 but is under different regulatory control.

Finally, the *spoZ-CD16671* mutant was confirmed via whole genome sequencing (see Methods). However, despite numerous attempts with various constructs and approaches, we were unable to complement the *spoZ-CD16671* mutant. In all, the regulatory portion of SpoZ is not well defined, and the regulation of the *spoZ-CD16671* locus is complex. Additional studies are needed to delineate the multifaceted molecular roles SpoZ plays in *C. difficile* spore formation.

## AgrD1 does not signal through the RgaS-RgaR two component system

The prototypical Agr signaling systems form a regulatory feedback loop, in which the quorum peptide signal (AgrD) activates the sensor kinase to initiate Agr-dependent regulation. As RgaS-RgaR directly activates *agrB1D1* transcription and a previously published *agrB1D1* mutant exhibits lower sporulation frequency [34], we hypothesized that the AgrD1 quorum-sensing peptide is recognized by RgaS to activate RgaR-dependent gene expression. To test this, we created an *agrB1D1* deletion mutant by allelic exchange (see Methods; **S9A Fig**). The *agrB1D1* mutant made fewer spores, similar to the *rgaS* and *rgaR* mutants, and expressing *agrB1D1* from an inducible promoter restored the sporulation frequency to greater than wild-type levels (**Fig 6A**).

We next asked whether the RgaS-RgaR-dependent gene expression was regulated by AgrD1. To answer this question, we assessed RgaR-dependent gene expression in the *agrB1D1* mutant. Surprisingly, qRT-PCR analysis revealed that transcript levels of *CD0587*, *CD2098*, *CD15111*, and *spoZ* were not dramatically reduced in the *agrB1D1* mutant (**Fig 6B**). Thus,

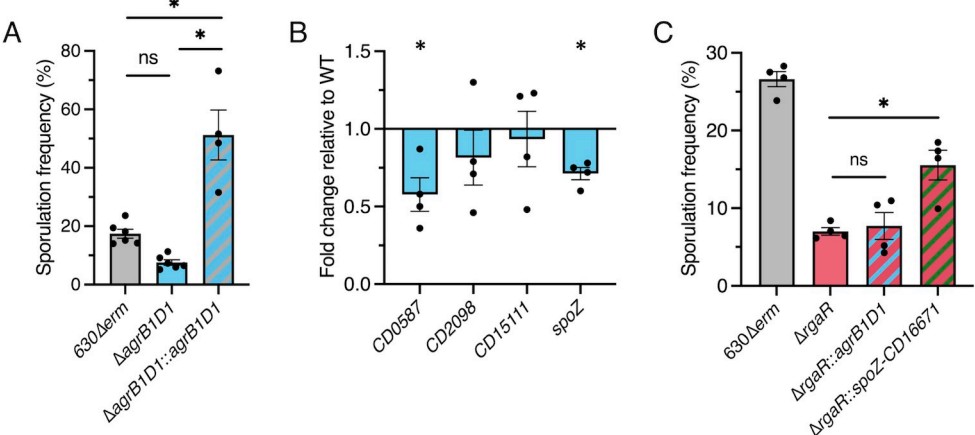

**Fig 6. AgrB1D1 does not activate RgaR-dependent gene expression and promotes sporulation upstream of SpoZ.**
(**A**) Ethanol-resistant spore formation of 630Δ*erm*, 630Δ*erm* Δ*agrB1D1* (MC2433), and 630Δ*erm* Δ*agrB1D1* Tn*916*::P*cprA-agrB1D1* (MC2524) grown on 70:30 agar at $H_{24}$. (**B**) Transcript levels of RgaR-dependent genes, *CD0587*, *CD2098*, *CD15111*, and *spoZ in* 630Δ*erm* and 630Δ*erm* Δ*agrB1D1* (MC2433) grown on 70:30 agar at $H_{12}$. (**C**) Ethanol-resistant spore formation of 630Δ*erm*, 630Δ*erm* Δ*rgaR* (MC2229), 630Δ*erm* Δ*rgaR* Tn*916*::P*cprA-agrB1D1* (MC2424), and 630Δ*erm* Δ*rgaR* Tn*916*::P*cprA-spoZ-CD16671* (MC2425) grown on 70:30 agar at $H_{24}$. The means and standard errors of the means for at least four biological replicates are shown. *, $P < 0.01$ by Student's *t*-test (B) or an one-way ANOVA followed by Tukey's multiple comparisons test (A, C).

RgaS is not the sensor for the AgrD1 peptide, as expression of RgaR-dependent genes is not affected by AgrD1. This unconnected decrease in sporulation for the *agrB1D1* mutant indicates that AgrB1D1 promotes sporulation through a mechanism that is independent of RgaSR.

Prior analyses of transcription in *agrB1*, *agrD1*, and *agrB1D1* mutants revealed that AgrD1 promotes transcription of early sporulation genes [34]. In contrast, disruption of *rgaR* most significantly impacted expression of late stage sporulation genes (**Table 1**), implying that SpoZ affects later stages of sporulation. As the *rgaR* mutant has significantly decreased transcript levels of both *agrB1D1* and *spoZ* (**Fig 3B**), we asked whether AgrD1 or SpoZ promote different stages of sporulation by performing an epistasis study in the *rgaR* mutant. We overexpressed either the *agrB1D1* or the *spoZ-CD16671* locus in the *rgaR* mutant and assessed sporulation frequency. We anticipated that if SpoZ is epistatic to AgrD1 (that is, if SpoZ functions downstream from AgrD1 in the sporulation regulatory pathway) that only *spoZ* overexpression will restore sporulation in the *rgaR* mutant. As expected, overexpression of *agrB1D1* did not increase sporulation of the *rgaR* mutant; however, overexpression of *spoZ-CD16671* in the *rgaR* mutant doubled spore formation (**Fig 6C**), demonstrating that *spoZ* is able to partially complement the low sporulation phenotype of the *rgaR* mutant. Since AgrD1 supports early sporulation, overexpression of *agrB1D1* cannot compensate for an *rgaR* mutant, as SpoZ must be required for later sporulation.

## The RgaS-RgaR regulatory network is conserved in the epidemic R20291 background

As the *agrB1D1* locus is highly conserved in almost all sequenced *C. difficile* genomes, we asked whether the RgaS-RgaR system is present and functions similarly in the epidemic ribotype 027 strain, R20291. The R20291 genome encodes highly conserved RgaS and RgaR orthologs, *CDR20291_0503* and *CDR20291_3113*. Null *rgaS* and *rgaR* mutants were created in the R20291 background (see Methods; **S9B Fig**). We assessed sporulation frequency and observed that the *rgaS* and *rgaR* mutants produced fewer spores (~1.6-fold and ~2.1-fold, respectively) than the R20291 parent strain (**Fig 7A**). Transcriptional analysis of the direct RgaR targets, *agrB1D1*, the *CD2098* ortholog (*CDR20291_2005*), the *CD15111* ortholog (unannotated in R20291; herein referred to as *CD15111^*), and *spoZ*, revealed significantly decreased transcript levels in the *rgaS* and *rgaR* mutants (**Fig 7B**), indicating that the RgaS-RgaR orthologs in R20291 function similarly as in 630Δ*erm*.

To further evaluate the RgaS-RgaR and AgrD1 regulatory circuitry in the R20291 background, we next asked whether the R20291 RgaS-RgaR system was responsive to the AgrD1 quorum-sensing peptide. To do this, we utilized CRISPRi to knockdown *agrB1* transcription, which significantly decreased transcript levels of both *agrB1* and *agrD1* (**Fig 7C**). We then assessed transcript levels of the direct RgaR targets, *CDR20291_2005*, *CD15111^*, and *spoZ*, which remained unchanged in the *agrB1D1* knockdown (**Fig 7C**). These data show that the RgaS-RgaR system also does not respond to AgrD1 in the R20291 background, indicating that this uncoupled signaling circuitry is conserved in diverse *C. difficile* strains.

To determine whether the regulatory sRNA SpoZ also impacts sporulation in R20291 through CD16671 (again, unannotated in R20291; herein referred to as *CD16671^*), we first examined *CD16671^* transcript levels in the R20291 *rgaS* and *rgaR* mutants. The R20291 *rgaS* and *rgaR* mutants had modestly increased *CD16671^* transcript levels (~3.8-fold and ~5.6-fold, respectively; **Fig 7D**), which was not as elevated as *CD16671* transcript levels were in the 630Δ*erm* background. As overexpression of *CD16671* inhibits spore formation in 630Δ*erm*, the modest increase in *CD16671^* transcript levels in R20291 likely explain why the R20291 *rgaS* and *rgaR* mutants do not exhibit as dramatic sporulation defects as the 630Δ*erm*

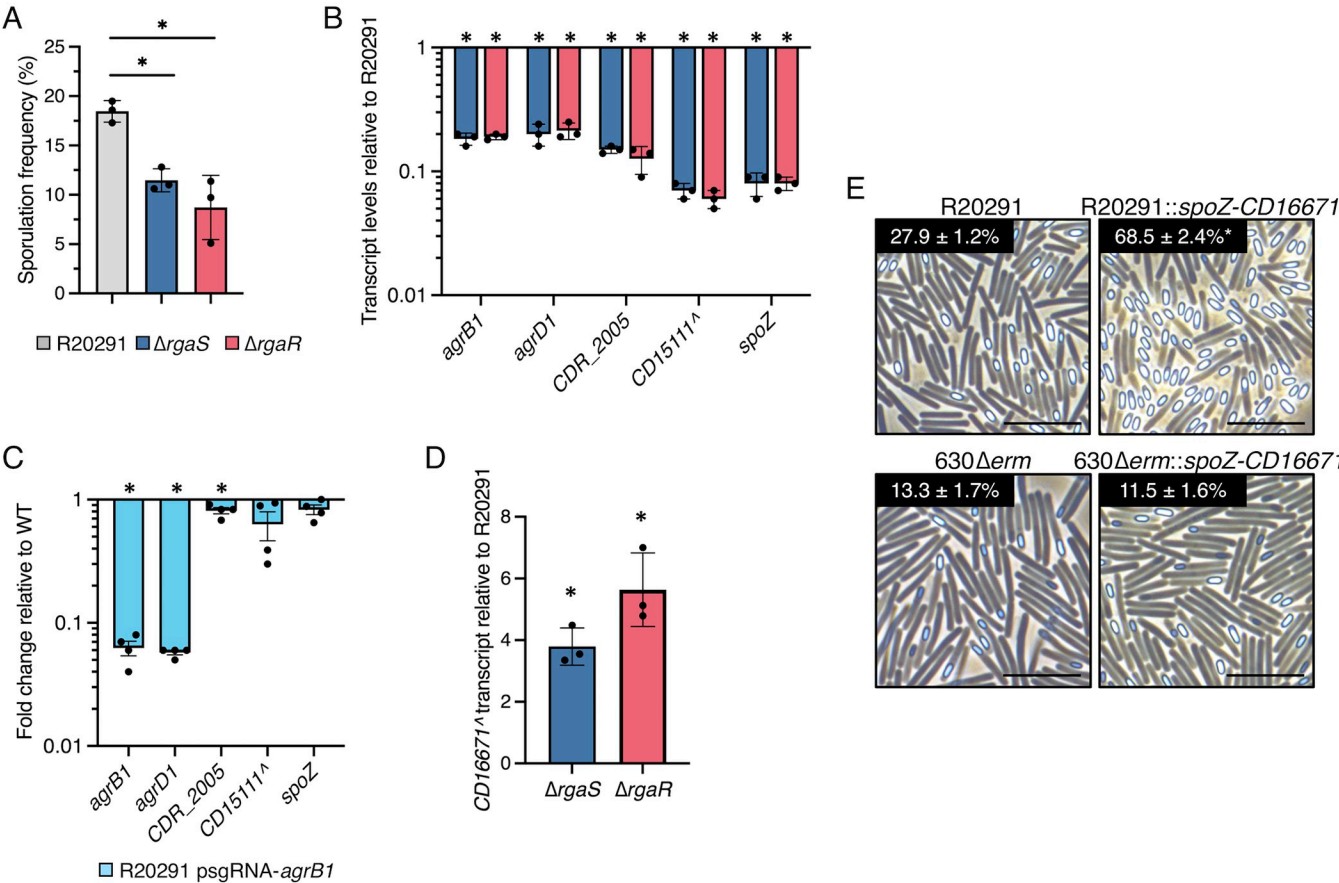

**Fig 7. The RgaS-RgaR-AgrB1D1-SpoZ regulatory network is conserved in R20291.** (**A**) Ethanol-resistant spore formation at H₂₄ and (**B**) qRT-PCR analyses of orthologous RgaR-dependent transcripts at H₁₂ in R20291, R20291 Δ*rgaS* (MC3278), and R20291 Δ*rgaR* (MC3279) grown on 70:30 agar. The means and standard deviations of three independent biological replicates are shown. (**C**) qRT-PCR analyses of orthologous RgaR-dependent transcripts in R20291 psgRNA-*neg* (MC2237) and R20291 psgRNA-*agrB1* (MC2534) grown on 70:30 agar at H₁₂. (**E**) Representative phase-contrast micrographs of R20291 and R20291 Tn*916*::P$_{cprA}$-*spoZ-CD16671* (MC2533) and 630Δ*erm* and 630Δ*erm* Tn*916*::P$_{cprA}$-*spoZ-CD16671* (MC2532) grown on 70:30 agar at H₂₄. Numbers below represent the means and standard error of the means of four independent biological replicates from ethanol-resistant sporulation assays at H₂₄. Scale bar denotes 10 μm. ^, The *CD15111* and *CD16671* loci in 630Δ*erm* are not annotated in the R20291 genome used for this study, so the 630Δ*erm* locus tag is used as a placeholder. *, $P < 0.01$ by one-way ANOVA followed by Dunnett's multiple comparisons test (A, B, D) or a Student's *t*-test (C, E).

*rgaS* and *rgaR* mutants (**Fig 7A** vs **Fig 2A**). Finally, we overexpressed the *spoZ-CD16671* locus in both the R20291 and 630Δ*erm* background via the Tn*916* transposon. Sporulation frequency in the R20291 *spoZ-CD16671* overexpression strain was increased 2.5-fold (**Figs 7E; S10A**), demonstrating that SpoZ significantly impacts spore formation in R20291 as well. Interestingly, overexpression of the *spoZ-CD16671* locus in the 630Δ*erm* background did not affect sporulation (**Figs 7E; S10B**). This difference in sporulation phenotypes observed between R20291 and 630Δ*erm* when *spoZ* is overexpressed may be because of the relative abundance of native *spoZ* is less in R20291 than in 630Δ*erm* (**Fig 5B**). Overall, these data indicate that the unusual RgaS-RgaR, AgrD1, and SpoZ regulatory circuitry uncovered in 630Δ*erm* are conserved and functional in the R20291 background.

## Discussion

Although the morphological changes throughout spore formation are conserved between clostridia and the well-studied bacilli, the regulatory pathways and molecular mechanisms that

govern sporulation have significantly diverged. Further, the mechanism of Spo0A phosphorylation, which activates sporulation, is unclear in *C. difficile*. While searching for sporulation factors that facilitate *C. difficile* sporulation initiation, we uncovered a complex regulatory circuit that promotes sporulation through an orthologous accessory gene regulator (Agr) system. We discovered a non-contiguous, cognate two-component system, comprised of the transmembrane histidine kinase RgaS and the response regulator RgaR. RgaS-RgaR directly activate transcription of several loci, including the *agrB1D1* operon and the regulatory small RNA, *spoZ*. We further demonstrated that the gene products of these loci, the quorum-sensing peptide, AgrD1, and the regulatory sRNA, SpoZ, impact *C. difficile* spore formation at different stages of sporulation. Finally, our investigation revealed that AgrD1 does not activate the RgaS-RgaR two-component system, breaking the conventional positive feedback featured in many characterized Agr systems.

To our knowledge, this is the first identification of an Agr-like two-component system that activates the transcription of the gene encoding the AIP precursor but does not serve as the receptor for the AIP. Classical Agr systems are activated through a positive feedback loop in which the histidine kinase (AgrC) autophosphorylates in response to the accumulation of AIP during high cell density conditions and subsequently transfers the phosphoryl group to the response regulator (AgrA), which in turn, activates transcription of target genes, including the genes that encode the Agr system components [49–52]. Our findings reveal an unusual Agr system in *C. difficile* that is not autoregulatory. Previous work demonstrated that an *agrD1* mutant does not impact *agrB1* transcript levels [34], corroborating that the AgrD1 peptide does not influence its own transcription. *C. perfringens* encodes a single *agrB1D1* locus, and the VirS-VirR two component system is the AIP receptor [53,39]; however, VirR does not activate *agrB1D1* transcription [54], also breaking the Agr autoregulatory loop. These examples suggest that clostridial Agr systems are not autoregulatory in general, although different mechanisms have evolved to unlink AIP accumulation to its synthesis. Considering that the *C. difficile* AgrD1 AIP is important for promoting spore formation independently of RgaS-RgaR, there must be another unidentified transmembrane sensor that serves as the AgrD1 receptor. Discovery of the *C. difficile* AgrD1 receptor will be important in elucidating the regulatory pathway through which AgrD1 promotes sporulation.

A striking feature of the RgaS-RgaR regulon is the direct transcriptional activation of a regulatory RNA. The regulon of Agr systems generally includes a gene encoding a small regulatory RNA. The most notable examples include the RNAIII regulatory RNA of *Staphylococcus aureus*, which is activated by AgrC-AgrA, and the VR-RNA, VirT, and VirU small regulatory RNAs in *C. perfringens*, which are activated by VirS-VirR [55–57]. These regulatory RNAs function as global regulators in both *S. aureus* and *C. perfringens* by influencing transcript stability and translational efficiency of target mRNAs [58–62]. Our discovery that RgaS-RgaR directly activates *spoZ* transcription suggests that *C. difficile* also coordinately regulates multiple targets through a single regulatory sRNA.

The function of SpoZ during *C. difficile* sporulation appears to be dependent on the accumulation of the hypothetical protein, CD16671. Low levels of *spoZ* transcript levels in the *rgaS* and *rgaR* mutants (~5% compared to the parent strain) led to low sporulation frequencies, while deletion or knockdown of *spoZ* resulted in hypersporulation. These sporulation phenotypes correlate with the abundance of the *CD16671* transcript: strains with low abundance of *spoZ* and high levels of *CD16671* exhibit low sporulation frequencies while those with low *spoZ* transcript levels and wild-type levels or lower of *CD16671* transcripts hypersporulate (**Table 2**). Altogether, these data suggest that SpoZ function is hierarchal, and the inhibitory effects of CD16671 mask the regulatory functions of other SpoZ targets. The inability to complement the *spoZ-CD16671* mutant may also be due to complex interplay between the *spoZ*

**Table 2. *CD16671* transcript levels correlate with the *C. difficile* sporulation phenotype in various *spoZ* mutants.**

| Genotype | *spoZ* transcript levels[a] | *CD16671* transcript levels[b] | Sporulation frequency[a] |
|---|---|---|---|
| 630Δ*erm* Δ*rgaR* | Low | High (20-fold) | Low |
| 630Δ*erm* p*CD16671* | WT | High (10-fold) | Low |
| 630Δ*erm* Δ*spoZ-CD16671* | ND | ND | High |
| 630Δ*erm* psgRNA-*spoZ* | Low | WT | High |

[a] Values are based on qRT-PCR transcript levels or ethanol-resistant sporulation assays compared to the parent strain. WT, wild-type; ND, not detected.

and *CD16671* transcripts. Additionally, a recent study employing Hfq RIL-seq identified three direct sRNA/mRNA interactions with SpoZ [63], indicating that SpoZ has multiple direct targets. These direct targets include the mRNAs of two genes, *CDIF630_01110* and *CDIF630_02206*), encoding hypothetical proteins, and another small regulatory RNA, *CDIF630nc_165*, which, in turn, binds to dozens of additional targets [63]. Intriguingly, only one of these direct SpoZ targets from the Hfq RIL-seq is encoded in R20291, perhaps offering another explanation of why overexpression of the *spoZ-CD16671* locus resulted in different sporulation phenotypes between the R20291 and 630Δ*erm* background (**Fig 7E**). Future directions will focus on the molecular mechanisms utilized by SpoZ to influence *C. difficile* sporulation.

Surprisingly, we found that our 630Δ*erm agrB1D1* mutants exhibited a variable sporulation phenotype that was selectable on plates. However, this variation was not noted with the previously published *agrB1D1* mutant, which was also constructed in the 630Δ*erm* background. Since both low and high sporulation phenotypes can be differentiated on plates after passaging a colony exhibiting a single colony morphology, we propose that there is a phase variation event occurring with higher frequency in the *agrB1D1* mutants that influences sporulation. Phase variation is a common strategy used by *C. difficile* to exhibit significant phenotypic heterogeneity [64–70], and there is a possibility that the Agr1 system may regulate a recombinase or other regulatory factor that influences the phase variation of sporulation.

The components of the RgaS-RgaR-SpoZ-AgrB1D1 regulatory circuitry are highly conserved in *C. difficile* genomes, suggesting these regulatory pathways are important in controlling earlier and later sporulation events across the species. The universal function of this system in *C. difficile* is further supported by data from the R20291 strain, which demonstrated conservation of the RgaS-RgaR regulon and sporulation phenotypes. In addition, bioinformatic analysis of the R20291 genome revealed the same RgaR targets, with no additional RgaR binding sites compared to the 630Δ*erm* strain.

Our working model of the Rga regulatory system (**Fig 8**) begins with RgaS autophosphorylation and phosphoryl group transfer to RgaR, which then directly activates both *agrB1D1* and *spoZ* transcription. The resulting pro-AgrD1 is processed and exported by AgrB1, accumulates extracellularly in high cell density conditions and promotes the initiation of sporulation through an unknown factor. The small regulatory RNA, SpoZ, targets one or more sRNA/mRNA(s) to influence late stage sporulation. One of these targets includes *CD16671*, whose expression and/or transcript stability is increased in the absence of SpoZ. CD16671 is proposed to inhibit late stage sporulation through an unknown mechanism. Finally, in the absence of *CD16671* accumulation, SpoZ affects at least one additional target (denoted as X) that likely positively impacts spore formation. In all, the RgaS-RgaR regulon integrates at multiple points within the sporulation pathway to control *C. difficile* spore formation. Understanding when RgaSR is most active and which signals activate (or inhibit) this system will aid in

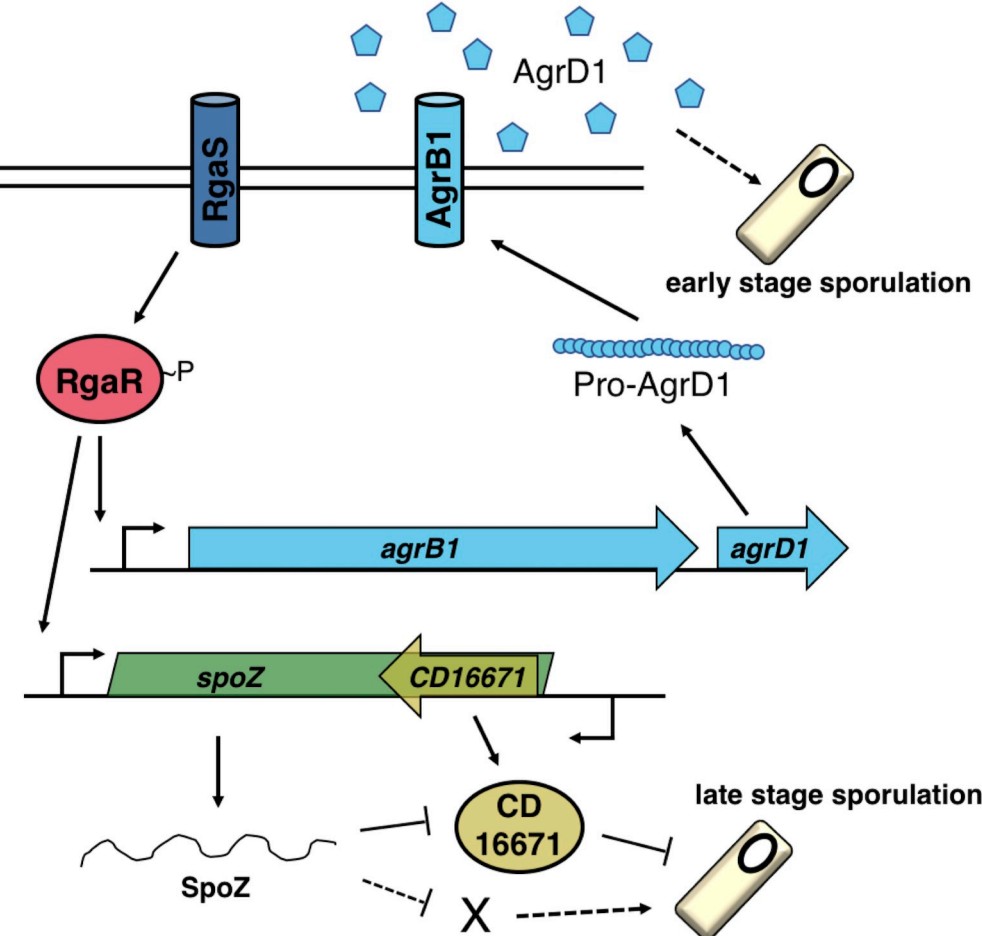

**Fig 8. The model of RgaSR influence on *C. difficile* sporulation.** The transmembrane sensor histidine kinase RgaS autophosphorylates in response to an unknown stimulus and transfers the phosphoryl group to the cytosolic response regulator RgaR. RgaR~P activates transcription of its regulon, including *agrB1D1* and *spoZ*. The gene products of the *agrB1D1* locus produce the extracellular quorum-sensing peptide, AgrD1, which promotes early stage sporulation through an unknown regulatory pathway. *spoZ* encodes a small regulatory RNA which influences *CD16671* transcript levels, and the presumed accumulation of CD16671, which inhibits late stage spore formation. In the absence of CD16671 accumulation, SpoZ negatively impacts sporulation via unidentified sRNA/mRNA targets. Known direct interactions are depicted with solid lines while unknown regulatory pathways are denoted with dashed lines.

understanding the environmental cues that promote *C. difficile* sporulation, permitting its survival in the environment and efficient transmission to new hosts.

## Methods

### Bacterial strains and growth conditions

The bacterial strains and plasmids constructed and used in this study are listed in **S2 Table**. Plasmid construction details are in **S3 Table**. *C. difficile* was routinely cultured in a Coy anaerobic chamber at 37˚C in an atmosphere of 10% $H_2$, 5% $CO_2$, and balanced with $N_2$, as previously described and grown in brain-heart infusion (BHI) broth supplemented with yeast extract [71–73]. For plasmid maintenance or selection in *C. difficile*, 2–10 µg/ml thiamphenicol, 100–500 µg/ml spectinomycin, or 2–5 µg/ml erythromycin were supplemented in the media as needed. Overnight cultures were supplemented with 0.1% taurocholate to allow

present spores to germinate and 0.2% fructose to prevent sporulation within the culture, as indicated [72,74]. *Escherichia coli* were grown at 37˚C in LB medium supplemented with 100 µg/ml ampicillin, 100 µg/ml spectinomycin, 300 µg/ml erythromycin, and/or 20 µg/ml chloramphenicol as needed, and *Bacillus subtilis* were grown at 37˚C in LB [75] supplemented with 100 µg/ml spectinomycin or 1 µg/ml erythromycin as needed. *B. subtilis* was also supplemented with 3.5 mM $KNO_3$ to support anaerobic growth. Kanamycin (100 µg/ml) was used to counterselect against *E. coli* and *B. subtilis* after conjugation with *C. difficile* [76].

## Strain construction

*C. difficile* 630 (GenBank accession no. NC_009089.1) and R20291 (GenBank accession no. NC_013316.1) were used as the templates for primer design, and *C. difficile* 630Δ*erm* and R20291 genomic DNA were used as the template for PCR amplification and mutant construction. The *C. difficile* 630Δ*erm* Δ*rgaS*, 630Δ*erm* Δ*rgaR*, 630Δ*erm* Δ*rgaS* Δ*rgaR*, 630Δ*erm* Δ*spoZ-CD16671*, 630Δ*erm* Δ*CD16671*, 630Δ*erm* Δ*agrB1D1*, R20291 Δ*rgaS*, and R20291 Δ*rgaR* strains were constructed using allele-coupled exchange using either the pMSR or pMSR0 pseudo-suicide vectors for the 630 and R20291 backgrounds, respectively [41]. Briefly, pMSR/pMSR0 vectors containing the targeted gene's 5′ and 3′ homology arms, with a selective antibiotic resistance cassette in between, were conjugated into the respective parent strain and were selected for growth on 10 µg/ml thiamphenicol and 100 µg/ml kanamycin. Colonies were then passaged to either 500 µg/ml spectinomycin or 5 µg/ml erythromycin and subsequently grown in 10 ml BHIS with the selective antibiotic and 100 µg/ml anhydrotetracycline (ATC) to induce the double recombination event. After ATC treatment, isolated colonies were screened for successful allelic exchange by PCR. Complementation strains for the *rgaS* and *rgaR* mutants were constructed by either expressing the wild-type or site-directed mutated *rgaS* or *rgaR* alleles via the conjugative transposon Tn*916*. *rgaR* mutants expressing either P*cprA-agrB1D1* or P*cprA-spoZ-CD16671* and the 630Δ*erm* and R20291 strains expressing P*cprA-spoZ-CD16671* were also constructed using the conjugative transposon Tn*916*. All strains were confirmed by PCR analysis, and most were further confirmed by whole genome sequencing (SeqCenter, Pittsburgh, PA; see below).

The Benchling CRISPR Guide RNA Design tool was used to create sgRNAs targeting *rgaS*, *rgaR*, *CD0587*, *CD2098*, *CD15111*, *spoZ*, and *agrB1* [35], which were generated by PCR and subsequently cloned into pMC1123. The details of vector construction are given in the supplemental material (see **S1 Text**). Plasmids pMC1249 and pMC1250 were synthesized by Genscript (Piscataway, NJ).

Finally, amino acid alignments were performed with Clustal Omega [77].

## Illumina whole genome sequencing

*C. difficile* strains were grown overnight in and harvested from 10 ml BHIS. After storing cell pellets overnight at -80˚C, genomic DNA was isolated using a modified Bust 'N' Grab protocol [26,78]. Contaminating RNA was removed by incubation with RNase A (Ambion) for 1 h at 37˚C. Library prep and Illumina sequencing were performed by SeqCenter (Pittsburgh, PA). Briefly, sample libraries were prepared using the Illumina DNA Prep Kit and IDT 10bp UDI indices, and sequenced on an Illumina NextSeq 2000, producing 2x151bp reads. Demultiplexing, quality control, and adapter trimming was performed with bcl-convert (v3.9.3; Illumina; https://support-docs.illumina.com/SW/BCL_Convert/Content/SW/FrontPages/BCL_Convert.htm). Using Geneious Prime v2022.3, the resulting reads were paired and trimmed using the BBDuk plug-in and subsequently mapped to the reference genome (630Δ*erm*; NC_009089.1). The Bowtie2 plug-in was used to search for the presence of SNPs and InDels

under default settings with a minimum variant frequency set at 0.85. Genome sequence files were deposited to the NCBI Sequence Read Archive (SRA) BioProject PRJNA986905.

## Sporulation assays and phase-contrast microscopy

*C. difficile* cultures were grown overnight in BHIS medium supplemented with 0.1% taurocholate, to promote germination of existing spores, and 0.2% fructose, to prevent sporulation during growth [72]. *C. difficile* cultures were subsequently diluted with fresh BHIS, grown to mid-exponential phase ($OD_{600} \cong 0.5$), and plated onto 70:30 agar supplemented with 2 μg/ml thiamphenicol and 1 μg/ml nisin as needed. After 24 h ($H_{24}$), ethanol-resistant sporulation assays were performed as previously described [14,79]. Briefly, cells were scraped from 70:30 agar and suspended in BHIS medium to an $OD_{600} \cong 1.0$. Total vegetative cells per milliliter were determined by immediately serially diluting suspended cells in BHIS, plating onto BHIS plates, and enumerating colony forming units (CFU) after at least 36 h of growth. Simultaneously, spores per milliliter were enumerated after a 15 min incubation in 28.5% ethanol by serially diluting in 1X PBS with 0.1% taurocholate, plating onto BHIS with 0.1% taurocholate, and counting after at least 36 h of growth. The sporulation frequency was calculated as the total number of spores divided by all viable cells (spores plus vegetative cells). A *spo0A* mutant was used as a negative sporulation control. Statistical significance was determined using a one-way ANOVA, followed by a Dunnett's or Tukey's multiple-comparison test, as indicated (GraphPad Prism 9.5.0). Phase-contrast microscopy was performed $H_{24}$, using the suspended cells from 70:30 agar, with a Ph3 oil immersion objective on a Nikon Eclipse Ci-L microscope. At least two fields of view were captured with a Ds-Fi2 camera. Results represent a minimum of three independent biological experiments.

## Enzyme-linked immunosorbent assay (ELISA)

*C. difficile* TcdA and TcdB from culture supernatants was quantified after 24 h growth of *C. difficile* strains in TY, pH 7.4, as previously described [80]. Supernatants, diluted in the provided dilution buffer, were assayed in technical duplicates using the tgcBIOMICS kit for simultaneous detection of both TcdA and TcdB, according to the manufacturer's instructions. The results represent the means and standard error of the means of four independent biological replicates, and statistical significance was assessed via a one-way ANOVA, followed by a Dunnett's multiple-comparison test (GraphPad Prism 9.5.0).

## Quantitative reverse transcription PCR analysis (qRT-PCR)

*C. difficile* strains were cultured on 70:30 agar as described above, and cells were harvested at $H_{12}$ (defined as 12 h after the cultures are applied to the plate) into 6 ml of 1:1:2 ethanol:acetone:$dH_2O$ solution and stored at -80°C. RNA was isolated and Dnase-I treated (Ambion), and cDNA was synthesized (Bioline) using random hexamers as previously described [16,18,81]. qRT-PCR analysis was performed in triplicate on 50 ng of cDNA using the SensiFAST SYBR & Fluorescein kit (Bioline) with a Roche Lightcycler 96. The results were analyzed by the comparative cycle threshold method [82], using the *rpoC* transcript as the normalizer, and are presented as the means and standard error of the means for a minimum of three independent biological replicates. An one-way ANOVA, followed by a Student's *t*-test or a Dunnett's multiple-comparison test as applicable (GraphPad Prism v9.5.0), was used to assess statistical significance.

## RNA sequencing (RNA-seq) analysis

*C. difficile* strains were cultured on 70:30 sporulation agar as described above and harvested at $H_{12}$ into 6 ml of 1:1:2 ethanol:acetone:dH$_2$O solution and stored at -80˚C. RNA was isolated and Dnase-I treated (Ambion). Samples were sent to Microbial Genomics Sequencing Center (MiGS; Pittsburgh, PA) where library preparation was performed using Illumina's Stranded Total RNA prep Ligation with Ribo-Zero Plus kit and 10bp IDT for Illumina indices. Sequencing was done on a NextSeq200 giving 2x50bp reads. Demultiplexing, quality control, and adapter trimming was performed with bcl-convert (v3.9.3; Illumina; see reference above). Using Geneious Prime v2022.2.2, the reads were mapped to the reference genome (630Δ*erm*; NC_009089.1). The expression levels were calculated and then subsequently compared using DESeq2 [83]. DESeq2 utilizes the Wald test to calculate *P* values which are then adjusted using the Benjamini-Hochberg test [83]. RNA-seq raw sequence reads were deposited to the NCBI Sequence Read Archive (SRA) BioProject PRJNA986905.

## Alkaline phosphatase activity assays

*C. difficile* strains were cultured on 70:30 sporulation agar as described above and harvested at $H_8$ (defined as eight hours after the cultures are applied to the plates). Alkaline phosphatase (AP) assays were performed as previously described [44] excepting the exclusion of chloroform. Technical duplicates were averaged and results represent the means and standard error of the means of three independent biological replicates. An one-way ANOVA, followed by a Tukey's multiple-comparison test (GraphPad Prism v9.5.0), was used to assess statistical significance.

## Northern blot analysis

Near-infrared fluorescent northern (irNorthern) blot analysis was performed as previously described [84]. Briefly, 2.5 nmol of a *spoZ* DNA probe (oMC3901), containing an internal azide modification (IDT, Coralville, IA), was fluorescently labeled with 50 nmol IRdye 800 CW (DBCO) in 1X phosphate buffered saline, pH 7.4 for 6 h at 25˚C. IRdye-labeled oligonucleotides were purified with Ampure XP beads (Beckman Coulter) as described [84], eluted in 50 µl nuclease-free dH$_2$O, and stored at -70˚C. For the irNorthern blot, 20 µg of RNA was diluted 1:1 with Glyoxal Load Dye (ThermoFisher Scientific), heated to 50˚C for 30 min, and electrophoresed on a 1.4% agarose gel. The RNA was transferred to a BrightStar-Plus positively charged nylon membrane (ThermoFisher Scientific) using downward capillary transfer, and the RNA was UV crosslinked to the membrane at 120 mJ/cm$^2$ using a Stratagene UV Stratalinker 1800. The membrane was prehybridized overnight at 37˚C in ULTRAhyb Hybridization Buffer (ThermoFisher Scientific), and hybridized with 1.3 nM IRdye-labeled DNA probe for 4 h at 37˚C. The membrane was washed once with Low Stringency Wash Solution 1 (ThermoFisher Scientific) for 10 min at room temperature, twice for 3 min at 37˚C. and subsequently visualized on a BioRad ChemiDoc MP. The size of the SpoZ transcript was estimated based on a standard curve calculated by the migration of the RNA Century Plus Marker (Ambion).

## Supporting information

**S1 Fig. Amino acid alignment of RgaS with the sporulation-associated phosphotransfer proteins, PtpA, PtpB, and PtpC.** The dimerization and histidine autophosphorylation domain (DHpt, or H-box) of the three sporulation-associated phosphotransfer proteins, PtpA (CD630_14920), PtpB (CD630_24920), and PtpC (CD630_15790) aligned with RgaS using

Clustal Omega.
(TIFF)

**S2 Fig. CRISPRi knockdown of rgaS and rgaR decreases rgaS and rgaR transcript levels.**
Transcript levels of rgaS (A) and rgaR (B) at H12 in 630Δerm psgRNA-neg (MC2065),
630Δerm psgRNA-rgaS (MC2066), and 630Δerm psgRNA-rgaR (MC2227) grown on 70:30
agar supplemented with 2 μg/ml thiamphenicol and 1 μg/m nisin, as indicated. The means and
standard deviation of three independent biological replicates are shown. *, P < 0.01 by a Stu-
dent's t-test.
(TIFF)

**S3 Fig. PCR confirmation of rgaS and rgaR mutants.** PCR verification of allelic replacement
of rgaS (CD0576) and rgaR (CD3255) with the spectinomycin (aad9) or erythromycin (ermB)
cassette, respectively in 630Δerm, 630Δerm ΔrgaS (MC2228), and 630Δerm ΔrgaR (MC2229).
Expected PCR products' sizes are: 3319 bp for the wildtype rgaS allele and 3078 bp for the
ΔrgaS::aad9 allele (primers oMC3123/3124); 2668 bp for the wildtype rgaR allele and 3436 bp
for the ΔrgaR::ermB allele (primers oMC3261/3262).
(TIFF)

**S4 Fig. Spo0A-dependent gene expression and toxin production are unaffected in C. diffi-
cile rgaS and rgaR mutants.** (A) qRT-PCR analysis of sigE in 630Δerm, 630Δerm ΔrgaS
(MC2228), and 630Δerm ΔrgaR (MC2229) grown on 70:30 agar at H12. (B) ELISA of TcdA
and TcdB present in the supernatant of 630Δerm, 630Δerm ΔrgaS (MC2228), and 630Δerm
ΔrgaR (MC2229) grown in TY medium, pH 7.4, at H24. The means and standard error of the
means of four independent biological replicates are shown. The statistical analysis used for
these data sets was a one-way ANOVA followed by Dunnett's multiple comparisons test.
(TIFF)

**S5 Fig. CRISPRi knockdown of RgaR-dependent genes specifically and significantly
decreases their transcript levels.** Transcript levels and of (A) CD0587, CD2098, CD15111,
and spoZ at H12 in 630Δerm expressing psgRNA-neg (MC2065) or CRISPRi targets for
CD0587 (MC2269), CD2098 (MC2270), CD15111 (MC2271), or spoZ (MC2272) grown on
70:30 agar supplemented with 2 μg/ml thiamphenicol and 1 μg/m nisin. The means and stan-
dard deviations of three independent biological replicates are shown. *, P < 0.0001 by a Stu-
dent's t-test.
(TIFF)

**S6 Fig. PCR validation of spoZ transcript length and predicted secondary structure.** (A)
Primers spanning the regions within the spoZ-CD16671 locus depicted in (B), were used to
amplify cDNA and RNA (minus reverse transcriptase negative control; -RT) from 630Δerm
grown on 70:30 agar at H12 and 630Δerm genomic DNA (gDNA). (C) Predicted secondary
structure of the 528 nt SpoZ sRNA as calculated by UNAFold. Shown is the structure with the
lowest free energy.
(TIFF)

**S7 Fig. PCR confirmation of spoZ-CD16671 and CD16671 mutants and qRT-PCR analysis
of transcript levels of CD16671 in a CD16671 overexpression strain.** (A, B) PCR verification
of allelic replacement of the spoZ-CD16671 locus with the spectinomycin (aad9) cassette in
630Δerm, 630Δerm ΔspoZ-CD16671 (MC2351; A) and 630Δerm ΔCD16671 (MC2363; B).
Expected PCR products' sizes are: 2165 bp for the wildtype spoZ-CD16671 allele, 2745 bp for
the ΔspoZ-CD16671::aad9 allele, and 3033 bp for the ΔCD16671::aad9 allele (primers
oMC3457/3458). (C) qRT-PCR of CD16671 transcript levels in 630Δerm pMC211 (MC282)

and 630Δerm pCD16671 (MC2561) grown on 70:30 agar at H7. The means and standard error of the mean is shown for five independent biological replicates. *, P < 0.001 by a Student's t-test.
(TIFF)

**S8 Fig. Transcript levels of CD19903, encoding a CD16671 ortholog, are not impacted by RgaR or SpoZ.** (A) Amino acid alignment of CD16671 and CD19903 using Clustal Omega (Sievers, et al. 2011 Mol Sys Biol. 7:539). (B) qRT-PCR analysis of CD19903 transcript levels in 630Δerm, 630Δerm ΔrgaR (MC2229), and 630Δerm ΔspoZ-CD16671 (MC2351) at H12 on 70:30 agar. *, P = 0.03 by one-way ANOVA followed by Dunnett's multiple comparisons test.
(TIFF)

**S9 Fig. PCR confirmation of 630Δerm agrB1D1 mutants and R20291 CDR20291_0503 (rgaS) and CDR20291_3113 mutants (rgaR).** (A) PCR verification of allelic replacement of the agrB1D1 locus with the spectinomycin cassette (aad9) in 630Δerm and 630Δerm ΔagrB1D1 Expected PCR sizes are 2585 bp for the wild-type allele and 2930 bp for the agrB1D1::aad9 allele (primers oMC3360/3361). (B) PCR verification of allelic replacement of rgaS and rgaR with the spectinomycin (aad9) cassette in R20291, R20291 ΔrgaS (MC2378), and R20291 ΔrgaR (MC2379). Expected PCR products' sizes are: 3215 bp for the wildtype rgaS allele and 2974 bp for the ΔrgaS::aad9 allele (primers oMC3124/3554); 2667 bp for the wildtype rgaR allele and 3056 bp for the ΔrgaR::aad9 allele (primers oMC3261/3262).
(TIFF)

**S10 Fig. Overexpression of spoZ differentially impacts sporulation in the R20291 background compared to 630Δerm.** Ethanol-resistant sporulation assays in (A) R20291 and R20291 Tn916::PcprA-spoZ-CD16671 (MC2533) and (B) 630Δerm and 630Δerm Tn916::PcprA-spoZ-CD16671 (MC2532) on 70:30 agar at H24. The means and standard errors of the means of four independent biological replicates are shown.
(TIFF)

**S1 Text DNA cloning and vector details.**
(DOCX)

**S1 Table RNA-seq of rgaR mutant compared to 630Δerm.**
(XLSX)

**S2 Table Bacterial strains and plasmids.**
(DOCX)

**S3 Table Oligonucleotides.**
(DOCX)

## Acknowledgments

We are grateful to the members of the McBride lab for their helpful suggestions, comments, and discussions throughout the course of this study. We are also thankful to Craig Ellermeier for the gift of pIA33. The content of this manuscript is solely the responsibility of the authors and does not necessarily reflect the official views of the National Institutes of Health.

## Author Contributions

**Conceptualization:** Adrianne N. Edwards, Shonna M. McBride.

**Data curation:** Adrianne N. Edwards, Shonna M. McBride.

**Formal analysis:** Adrianne N. Edwards.

**Funding acquisition:** Shonna M. McBride.

**Investigation:** Adrianne N. Edwards, Shonna M. McBride.

**Resources:** Shonna M. McBride.

**Visualization:** Adrianne N. Edwards, Shonna M. McBride.

**Writing – original draft:** Adrianne N. Edwards.

**Writing – review & editing:** Shonna M. McBride.

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
