## [Decision Letter · Decision Letter 0]

2 Aug 2023

Dear Dr Edwards,

Thank you very much for submitting your Research Article entitled 'The RgaS-RgaR two-component system promotes Clostridioides difficile sporulation through a small RNA and the Agr1 system' to PLOS Genetics.

The manuscript was fully evaluated at the editorial level and by independent peer reviewers. The reviewers appreciated the attention to an important problem, but raised some substantial concerns about the current manuscript. Based on the reviews, we will not be able to accept this version of the manuscript, but we would be willing to review a much-revised version. We cannot, of course, promise publication at that time.

If you decide to revise the manuscript for further consideration at PLOS Genetics, please aim to resubmit within the next 60 days, unless it will take extra time to address the concerns of the reviewers, in which case we would appreciate an expected resubmission date by email to plosgenetics@plos.org.

We are sorry that we cannot be more positive about your manuscript at this stage. Please do not hesitate to contact us if you have any concerns or questions.

Yours sincerely,

Aimee Shen

Academic Editor

PLOS Genetics

Lotte Søgaard-Andersen

Section Editor

PLOS Genetics

The Reviewers were in agreement that your discovery of a new two-component system, RgaS/RgaR in Clostridioides difficile that modulates sporulation levels through an Agr quorum-sensing system and a novel small regulatory RNA, SrsR, reveals an additional layer to the regulation of this important developmental process. While they found the manuscript to be well written and the experiments rigorously conducted, they were in agreement that additional mechanistic insight into how RgaS/RgaR promote sporulation would increase the impact of the current study. Reviewer 3 would like additional data addressing whether RgaR/RgaS impact Spo0A activity/activation. Reviewer 2 suggested some in silico analyses to potentially gain insight into how SrsR promotes sporulation (they also suggest renaming SrsR, since two small regulatory RNAs were recently shown to alter Spo0A levels and thus regulate sporulation) and would like clarification on the position of the sRNA gene with CD16671. Conducting some northern blots or RNAse protection assays to gain insight into the nature of the specific transcripts associated with srsR and CD16671 might help refine the authors’ model for how SrsR modulates sporulation. Related to the authors’ working model, SrsR appears to antagonize the levels of CD16671 transcript, so Reviewer 2 suggests over-expressing CD16671 to recapitulate the reduced sporulation mutant of rgaR/rgaS mutants.

If the authors’ data support a role for RgaS/RgaR in regulating sporulation primarily at later stages, the authors may want to consider focusing a revised manuscript more on RgaS/RgaR’s regulation of SrsR function rather than on the Agr system. This may help the reader follow the complex regulatory picture uncovered by the authors more clearly. Using a more precise nomenclature when referring to mutant strains would also improve readability, since mutants are often shown in figures without a ∆ symbol when they are knock-outs, or only a part of the mutation is referred to (e.g mutations that impact both srsR-CD16671 should be described as such rather than sometimes as an srsR deletion).

Reviewer's Responses to Questions

**Comments to the Authors:**

Reviewer #1: Review has been uploaded as an attachment

Reviewer #2: Clostridioides difficile is a major gastrointestinal pathogen. The ability of this bacterium to produce spores is important for its survival in the environment, persistence in the gut, and transmission from the environment to the gut when causing disease. Despite this importance, the regulation of C. difficile sporulation remains incompletely understood at present.

This manuscript helps to address that knowledge gap by identifying orphan response regulator RgaR as a functional partner of the orphan histidine kinase RgaS in a two component regulatory system that positively-regulates several genes, including two involved in controlling sporulation. One of those two RgaS/RgaR-regulated genes encodes AgrD1; the current study confirms previous reports that this quorum sensing peptide positively regulates C. difficile sporulation and demonstrates that this regulation occurs early in the sporulation pathway. The other RgaS/RgaR-regulated gene affecting sporulation encodes SrsR, which is a small regulatory RNA. SrsR is shown to positively affect sporulation at later stages in the process.

Overall, the study did not accomplish their original goal (to identify the kinase phosphorylating Spo0A). However, this study (with a few exceptions noted below) is scientifically sound and does identify a new two component regulatory system in C. difficile that helps to regulate sporulation. It also sheds some new insights into the regulation of C. difficile sporulation. In particular, the discovery of a new small RNA involved in regulating sporulation is interesting, although precisely how this sRNA affects sporulation will require further study. There are also some intriguing findings that identify unique aspects of the C. difficile Agr quorum sensing system.

Specific comments:

1) Line 131: please briefly describe this limited similarity between the rgaS gene and other sporulation-associated phosphotransfer proteins.

2) In Fig. 1A, it might help clarity if the x axis indicated that the right panel is 70:30 + nisin.

3) In Fig. 1B, was this identity/similarity of RgaS to VirS and AgrCs distributed throughout the proteins or did it cluster preferentially to either the N-terminal membrane sensor domain or the C-terminal cytoplasmic phosphorylation domain?

4) Line 168: it is interesting that the screen identified genes encoding two VirR orthologs, RgaR and RgbR. The rationale for focusing on RgaR is clear and adding RgbR studies to the current manuscript would be unreasonable; however, the Discussion might revisit RgbR and mention the need for future studies of this response regulator. Might RgbR also be involved in regulating sporulation in C. difficile?

5) Relative to wild-type, did the RgaS and RgaR mutants (and other mutants made in this study) show any differences in their vegetative growth?

6) In Fig 2A and elsewhere, the x axis labels are sometimes challenging to follow. Suggest changing rgaS to (delta)rgaS for clarity and consistency with the Figure Legend.

7) In the Fig 2 legend, bold (C) for consistency with (A) and (B); on line 202, replace (B) with (C).

8) Table 1: Agree that the data suggest that rgaR mutant preferentially affects late sporulation gene expression. However,

a. Please indicate what fold-change was considered a “significant” regulatory effect in this RNA-seq experiment?

b. It is interesting that the only upregulated genes in the rgaR mutant were Spo0A dependent. Do the authors have any thoughts about this?

c. The RNA-seq data indicates that spo0A transcription was affected in the rgaR mutant- does this mutant have altered Spo0A or Spo0A-P levels?

d. It is mentioned on line 375 that a prior study showed that AgrD1 promotes transcription of early sporulation genes. Does this include spo0A? If that is known, then does the AgrBD mutant, which shows reduced sporulation due to an effect early in sporulation, exhibit less Spo0A and Spo0A phosphorylation than wild-type? Answering this question is perhaps slightly off-topic for the current study but might begin to shed some mechanistic insights into the role of AgrD in regulating the early steps of C. difficile sporulation.

9) Line 294: add “expression” after CD16671

10) Line 302: add a reference after “interactions”.

11) Is Fig. 5B ever addressed in the text of the results?

12) On print-outs it is sometimes hard to distinguish asterisks from solid dots in figures. Consider making these symbols slightly larger.

13) In the Fig. 8 legend describe the meaning of “X” and “Y”; presumably these refer to the abundance of SrsR RNA?

14) References need some work. Please double check.

a. The style is inconsistent. Specifically, in titles of some references the first letter of every word is capitalized, while in other references the first letter of every word in the title is capitalized. PLoS Genetics presumably has a reference style and this should be adhered to. If not, then pick one style and be consistent.

b. Some references are incomplete. Fpr example, reference 12 lacks page numbers.

c. Reference 20: please correct the title of this reference. There are some extraneous symbols present.

Reviewer #3: This work from Edwards and McBride provides further insight into the unique complexities of regulatory networks controlling C. difficile sporulation. Leveraging sequence similarity to other clostridial regulators of Spo0A phosphorylation they examined a cognate histidine kinase and response regulator. RgaS-RgaR was discovered to control late-stage sporulation. Suppressing expression of RgaS and/or RgaR reduced sporulation efficiency, which was recovered by complementation. They further connected this to control of a small RNA, SrsR, but were unable to dissect this regulatory RNA’s mechanism of action. They further eliminate regulation by AgrD1 yet show connections to expression of the Agr system. Finally, they show the system is conserved in other strains of C. difficile. Overall, the paper is well-written, and the experiments are rigorous. Although there is more work to be done, further investigation into SrsR will be a major undertaking. As such, I’m comfortable with leaving that portion of the study open for interpretation.

The authors are quick to conclude that RgaS-RgaR has little influence on Spo0A. What is the C. difficile Spo0A regulon? How has that been defined and reported? I’m familiar with the previous work from the Shen group, but my recollection is that study focused on sigma factors. I don’t recall a C. difficile Spo0A regulon reported in the literature and the authors don’t reference one.

Is it possible the authors have discovered a more refined Spo0A regulon subject to upstream control by RgaS and/or RgaS-RgaR? One might posit RgaS phosphorylates Spo0A and transcription of SrsR tempers the rate of sporulation, maybe as a checkpoint control in sporulation. Thus, when RgaS is absent Spo0A is not activated and sporulation is reduced, regardless of SrsR activity. When srsR is deleted, Spo0A might still be phosphorylated thereby explaining the hypersporulation phenotype. This could explain the “internally inconsistent” data mentioned in the discussion. Unfortunately, the authors didn’t check for changes in Spo0A phosphorylation so it's difficult to know if this model is within the realm of possibilities. The authors should examine Spo0A phosphorylation in their knockdown strains.

**Have all data underlying the figures and results presented in the manuscript been provided?**

Reviewer #1: Yes

Reviewer #2: Yes

Reviewer #3: Yes

PLOS authors have the option to publish the peer review history of their article (what does this mean?). If published, this will include your full peer review and any attached files.

Reviewer #1: No

Reviewer #2: No

Reviewer #3: No

---

## [Decision Letter · Decision Letter 1]

2 Oct 2023

Dear Dr Edwards,

We are pleased to inform you that your manuscript entitled "The RgaS-RgaR two-component system promotes Clostridioides difficile sporulation through a small RNA and the Agr1 system" has been editorially accepted for publication in PLOS Genetics. Congratulations!

Yours sincerely,

Aimee Shen

Academic Editor

PLOS Genetics

Lotte Søgaard-Andersen

Section Editor

PLOS Genetics

Comments from the reviewers (if applicable):

We thank the authors for their attention to the Reviewers comments. The additional experiments enhance an already excellent story.

Reviewer's Responses to Questions

**Comments to the Authors:**

Reviewer #1: I would like to thank the authors for addressing all my comments so comprehensively. I am happy that I could contribute some helpful suggestions. I have no further concerns and would be happy to see this work published.

Reviewer #3: The authors have addressed each of my original concerns with new experiments and significant modifications to the first submission.

**Have all data underlying the figures and results presented in the manuscript been provided?**

Reviewer #1: Yes

Reviewer #3: Yes

PLOS authors have the option to publish the peer review history of their article (what does this mean?). If published, this will include your full peer review and any attached files.

Reviewer #1: No

Reviewer #3: No

**Data Deposition**

http://datadryad.org/submit?journalID=pgenetics&manu=PGENETICS-D-23-00713R1

**Press Queries**

---

## [Editor Report · Acceptance letter]

11 Oct 2023

PGENETICS-D-23-00713R1 

The RgaS-RgaR two-component system promotes *Clostridioides difficile* sporulation through a small RNA and the Agr1 system 

Dear Dr Edwards, 

We are pleased to inform you that your manuscript entitled "The RgaS-RgaR two-component system promotes *Clostridioides difficile* sporulation through a small RNA and the Agr1 system" has been formally accepted for publication in PLOS Genetics! Your manuscript is now with our production department and you will be notified of the publication date in due course.

With kind regards,

Zsofi Zombor

PLOS Genetics

On behalf of:
